

# Carbon capture, photosynthesis, and leaf gas exchange of shade tree species and Arabica coffee varieties in coffee agroforestry systems in Veracruz state, Mexico

Daniel Cabrera-Santos[1], Patricia Dávila[2], Isela Rodríguez-Arévalo[2], Anabel Ruiz-Flores[1], Josefina Vázquez-Medrano[1], Salvador Sampayo-Maldonado[1], Cesar Ordoñez-Salanueva[1], Maraeva Gianella[3], Elizabeth Bell[3], María Toledo-Garibaldi[4], Robert Manson[4], Flor G. Vázquez-Corzas[5], Jazmin Cobos-Silva[5], Cesar Mateo Flores Ortiz[1,6] and Tiziana Ulian[3,7]

[1] Laboratorio de Fisiología Vegetal, Unidad de Biotecnología y Prototipos (UBIPRO), FES Iztacala, Universidad Nacional Autónoma de México, Tlalnepantla, Estado de México, Mexico

[2] Laboratorio de Recursos Naturales, Unidad de Biotecnología y Prototipos (UBIPRO), FES Iztacala, Universidad Nacional Autónoma de México, Tlalnepantla, Estado de México, Mexico

[3] Royal Botanic Gardens Kew, Wakehurst, Ardingly, United Kingdom

[4] Red de Ecología Funcional, Instituto de Ecología, A.C., Xalapa, Veracruz, Mexico

[5] Pronatura Veracruz AC, Coatepec, Veracruz, Mexico

[6] Laboratorio Nacional en Salud, Universidad Nacional Autónoma de México, Tlalnepantla, Estado de México, Mexico

[7] Department of Life Sciences and Systems Biology, University of Turin, Turin, Italy

Corresponding author
Cesar Mateo Flores Ortiz, cmflores@unam.mx

## ABSTRACT

Agroforestry systems with native vegetation enhance climate adaptation and mitigation by improving coffee farm resilience, carbon storage, and income diversification. Seven native tree species were pre-selected as shade providers for Veracruz coffee agroforestry systems based on ecological, cultural, and economic criteria. The present study evaluated their physiological performance through above-ground biomass, carbon stocks, and *in-situ* chlorophyll fluorescence and gas exchange measurements under controlled light and temperature conditions. Five *Coffea arabica* varieties were also assessed under these shade canopies using the same leaf-level parameters, and leaf nitrogen and moisture content. *Erythrina americana* and *Persea schiedeana* had the highest carbon sequestration per tree. *E. americana* showed the highest water-use efficiency, whereas *P. schiedeana* showed the lowest transpiration and stomatal conductance, indicating a water-saving strategy via stomatal restriction. These traits reflect their ecological adaptations to shade and microclimate conditions in agroforestry systems. *Inga inicuil* achieved the highest carbon capture per hectare due to high tree density, despite lower individual performance. Species-specific strategies were identified: *Psidium guajava* and *P. schiedeana* exhibited high transpiration but limited carbon gain. *E. americana* and *Inga punctata* formed a drought-resilient group, having a high carbon assimilation and low water loss. Intermediate species (*Heliocarpus appendiculatus*, *Inga vera*, *I. inicuil*) balanced moderate $CO_2$ assimilation rates with adaptable stomatal response. Photochemical efficiency remained stable across species.

Shaded *Coffea arabica* var. Oro Azteca had significantly higher leaf nitrogen, moisture, and water-use efficiency than unshaded ones. These differences coincided with lower PAR under shade, aligning with known variations in shaded versus unshaded coffee plants. Principal component analysis showed that PC1 correlated strongly with stomatal conductance and transpiration, driven by *P. guajava* and *P. schiedeana*. PC2 showed a carbon economy trade-off between $CO_2$ assimilation and internal concentration, dominated by *E. americana*. Collectively, these components highlight stomatal regulation and carbon management as adaptive strategies. Coffee PCA revealed contrasting water-use strategies: PC1 showed inverse stomatal regulation (especially in shaded varieties), and PC2 an energy allocation trade-off between photochemical efficiency and carbon assimilation, with shaded plants maintaining stable $CO_2$ assimilation regarding unshaded ones. These results demonstrate notable interspecific variation in carbon storage, water-use efficiency, and light conditions among shade trees, offering empirical support for species selection in Veracruz coffee agroforestry.

# INTRODUCTION

Coffee is one of the most widely traded and consumed agricultural commodities worldwide (*FAO, 2022*) with 70% produced in Latin America (*Baffes, Lewin & Varangis, 2005*). Coffee production during 2023 was estimated at 10.1 million tonnes, with an expected growth rate of 5.8% by 2024 (*FAO, 2022*; *ICO, 2023*). During 2017–2022, 70% of the world's coffee production was exported from producing nations to other countries, generating USD 19 billion in revenue and employing 125 million people globally (*Panhuysen & Pierrot, 2020*; *Fairtrade Foundation, 2022*).

The coffee industry was valued at US$132.13 billion in 2024, and globally, there are 12.5 million coffee farms, many located in high-biodiversity zones and managed by smallholders (*Donald, 2004*; *Panhuysen & Pierrot, 2020*; *Mordor Intelligence, 2024*). In Mexico, coffee is produced on approximately 580,000 ha by 481,000 farmers (*Ellis et al., 2010*; *Harvey et al., 2021*; *USDA FAS, 2023*). Approximately 86% of production in the country comes from *Coffea arabica* L. cultivation in shade polycultures that support biodiversity and provide ecosystem services (*Beer et al., 1998*; *Moguel & Toledo, 1999*; *Dávalos-Sotelo, Morato & Martínez-Pinillos-Cueto, 2008*; *Jha et al., 2011*; *Toledo & Moguel, 2012*).

Mexican coffee production, which has historically resisted intensification (*Rice, 2008*), now faces climate-induced land use changes threatening biodiversity (*Toledo & Moguel, 2012*) and ecosystem services (*Beer et al., 1998*). Coffee is highly climate-sensitive, with projected range reductions due to shifting temperature, humidity, and rainfall patterns (*Bunn et al., 2015*; *Pham et al., 2019*; *Bilen et al., 2023*). Agroforestry systems mitigate these impacts through shade-regulated microclimates, carbon capture (*Verchot et al., 2007*;
*Noponen et al., 2013*; *Rahn et al., 2014*; *Jawo, Kyereh & Lojka, 2022*; *Terasaki Hart et al., 2023*), and soil conservation (*Segura, Kanninen & Suárez, 2006*; *Lin, 2007*; *Lin, 2010*; *Siles et al., 2012*; *Notaro et al., 2014*; *Ehrenbergerová, Šenfeldr & Habrova, 2018*), while diversifying income *via* biomass and other attributes of native trees (*Acevedo et al., 1992*; *Jose, 2009*; *Häger, 2012*; *Noponen et al., 2013*; *Gross et al., 2022*). Optimal shade species should be native to enhance soil fertility and pest control without compromising biodiversity (*Gill & Prasad, 2000*; *Reigosa et al., 2000*; *Gliessman, 2015*). The Mexican state of Veracruz exhibits exceptional biodiversity (*Rzedowski, 1978*; *Estrada-Contreras et al., 2015*; *Tellez et al., 2020*), making its coffee agroforestry systems particularly valuable for studying carbon sequestration. These systems produce 23% of Mexico's coffee (*Nestel, 1995*) and can store 73.27 Mg C ha$^{-1}$ in coffee-tall tree combinations (*Ortiz-Ceballos et al., 2020*), retaining 91.2% of forest cover while supporting biodiversity comparable to secondary forests (*Dávalos-Sotelo, Morato & Martínez-Pinillos-Cueto, 2008*; *Vizcaíno-Bravo, Williams-Linera & Asbjornsen, 2020*). This evidence highlights their dual role in climate mitigation and conservation.

Plant photosynthesis is pivotal for $CO_2$ mitigation through carbon assimilation into biomass (*Fini et al., 2023*; *Jin et al., 2023*). Accurate carbon sequestration estimates rely on above-ground biomass measurements using allometric equations, yielding realistic estimates of assimilated carbon (C) into forest biomass and, consequently, of carbon cycling in ecosystems (*Liang & Wang, 2020*; *Araza et al., 2022*). Physiological assessments, particularly leaf gas exchange and chlorophyll fluorescence parameters, offer critical insights into performance and plant acclimatation to environmental changes (*Genty, Briantais & Baker, 1989*; *Sakshaug et al., 1998*; *Roháček, 2002*; *Baker & Rosenqvist, 2004*; *Strasser, Tsimilli-Michael & Srivastava, 2004*; *Baker, 2008*), enhancing the comprehension of ecosystem carbon dynamics.

Shade trees in agroforestry systems critically regulate microclimatic conditions, affecting the physiological responses of coffee plants and associated tree species. Evidence shows that shade reduces air temperatures by 1–5 °C, alleviating heat stress and maintaining optimal leaf temperatures (20–24 °C) for photosynthesis (*Vaast et al., 2005*; *Lara-Estrada, Rasche & Schneider, 2023*). These conditions improve chlorophyll fluorescence parameters including Fv/Fm (quantum yield of PSII) by reducing photodamage under excessive irradiance (*Rodríguez-López et al., 2014*). However, shade-adapted leaves typically exhibit lower heat tolerance than sun leaves, as observed in tropical trees like *Inga spectabilis* (Vahl) Willd., with shade leaves exhibiting a reduced threshold for PSII dysfunction (*Slot et al., 2019*). Light availability substantially mediates leaf gas exchange in coffee plants. Shade levels of 30–50% optimise photosynthetic rates through balanced irradiance and photoprotection, whereas excessive shade (>60%) can reduce light-saturated photosynthesis and yield by 10–30% (*DaMatta, 2004*; *Haggar et al., 2011*; *Isaac et al., 2024*). Notably, shade-tree traits such as layered canopies or high leaf nitrogen enhance nutrient cycling and light diffusion, further modifying these physiological responses (*Sauvadet et al., 2019*; *Isaac et al., 2024*).

Under climate change scenarios, water availability represents a critical challenge for agricultural and natural ecosystems. Water-use efficiency (WUE), calculated in our study as the ratio of $CO_2$ assimilation to transpiration ($CO_2/H_2O$) during gas exchange (*Kirkham,*

*2005*), reflects short-term trade-offs between carbon gain and water loss under controlled conditions and couples plant productivity to water management (*Bhattacharya, 2019*; *Hatfield & Dold, 2019*). WUE exhibits interspecific variation according to plant functional traits and environmental conditions (*Chaves, Osório & Pereira, 2004*; *McCarthy, Pataki & Jenerette, 2011*). Agroforestry shade enhances WUE through increased ambient humidity (10–20%) and reducing stomatal conductance, although shaded coffee plants frequently exhibit higher mass-specific transpiration due to morphological adaptations like thinner leaves (*Lin, 2010*; *Sarmiento-Soler et al., 2019*; *de Carvalho et al., 2021*; *Koutouleas et al., 2022*). Collectively, these findings demonstrate shade's role in microclimate stabilisation and photosynthetic optimisation, while highlighting knowledge gaps regarding species-specific physiological trade-offs.

Therefore, the present work examines the ecophysiological characteristics of seven previously selected shade tree species and five *C. arabica* varieties that make up agroforestry systems in central Veracruz, Mexico. For this task, *in-situ* dendrometric measurements of aboveground biomass, chlorophyll fluorescence, gas exchange parameters, and nitrogen and moisture content of leaves located at the understory layer of the selected shade tree species and coffee varieties were conducted under controlled temperature and photosynthetic photon flux density (PPFD) conditions.

We hypothesised that the physiological traits of these seven tree species identified by *Flores-Ortiz et al. (2025)* and their interactions with coffee varieties growing in shaded environments enhance carbon capture capacity while improving agroforestry system resilience under climate change. These effects are mediated through stress reduction *via* shading and optimised resource-use efficiency. We expect that this information will help decision-making during the selection and management of suitable shade trees based on their ecophysiological characteristics and carbon capture potential in shade coffee agroforestry systems.

## MATERIALS AND METHODS

### Selection of shade tree species and Arabica coffee varieties in agroforestry systems

Seven native tree species were selected from an initial screening of 50 conducted by *Flores-Ortiz et al. (2025)* based on conservation status, growth rate, and agroecological utility for coffee production. Species used for firewood or fuel were excluded, along with cultivated *Persea americana* Mill. (Hass avocado) due to its intensive domestication. For the selected shade trees species: *Inga inicuil* Schltdl. and Cham. Ex G. Don (*Ii*); *Inga vera* Willd. (*Iv*); *Inga punctata* Willd. (*Ip*); *Erythrina americana* Mill. (*Ea*); *Psidium guajava* L. (*Pg*); *Persea schiedeana* Turcz. (*Ps*), and *Heliocarpus appendiculatus* Nees (*Ha*), dendrometric, chlorophyll fluorescence, and gas exchange characteristics were measured in trees aged ≤30 years.

Due to the heterogeneity of *C. arabica* varieties found in the coffee farms in the region, chlorophyll fluorescence and gas exchange parameters measured were focused on Oro Azteca (Oa), Garnica (G), Costa Rica 95 (Cr), Tipica (T), and Catuai amarillo (Ca)

plants aged 4–6 years. To enable comparison, identical physiological measurements were conducted for Oro Azteca variety plants under unshaded conditions.

This selection aimed to represent a functional diversity of native species with potential for enhancing ecosystem services in coffee farms.

### Study area

The shade tree species and coffee varieties studied make up agroforestry systems in a traditional polyculture configuration and in the intermediate secondary succession stage. Coffee plants in unshaded conditions were part of an unshaded monoculture system. The tree density, considering trees between 10–15 m in height and >5 cm in diameter at breast height (DBH, at 1.30 m) for this type of coffee agroforestry system and coffee region, has been reported at ∼1,000 trees ha$^{-1}$ (*López-Gómez, Williams-Linera & Manson, 2008*; *Williams-Linera & Lorea, 2009*).

Study sites were located in central Veracruz on shaded and unshaded coffee farms in the municipalities of Teocelo (19°23′36″N, 96°59′9.4″W, at an elevation of 1,117 m a. s. l., average air temperature of 23.43 ± 0.37 °C and 70.4 ± 5.95% of relative humidity, or RH) and Xico (19°25′23.5″N, 96°55′42.6″W, at an elevation of 1,053 m a. s. l., average air temperature of 27.53 ± 0.86 °C and 60.23 ± 4.16% of RH for shaded and unshaded conditions, respectively (Fig. 1; Map created using the Free and Open Source QGIS).

To ensure environmental homogeneity, measurements were taken during two consecutive cool-season months (October–November 2022). This period exhibited typical climatic conditions for 2016–2022 trends, with precipitation, temperature, and cloud cover within expected ranges (*Weather Spark, 2024*). The cool season brought average maxima of 24 °C (Teocelo) and 26 °C (Xico), alongside ≥1 mm/h precipitation and 80% cloud cover. Historical records show Teocelo's temperature extremes (21–35 °C, 1945–2020), while Xico ranged from 9–12 °C (minima) to 21–22 °C (maxima, 1966–2023) (Servicio Meteorológico Nacional, https://smn.conagua.gob.mx/es/climatologia/informacion-climatologica/normales-climatologicas-por-estado?estado=ver (accessed 17 November 2025)). Both sites share comparable annual precipitation (1,847 and 2,091.8 mm for Xico and Teocelo, respectively; period 1991–2020) and stable conditions due to their proximity.

### Sampling strategy

The sampling design was structured to capture intra- and inter-specific ecophysiological variability. For each of the seven shade tree species, three mature and reproductive individuals were randomly selected (individuals = three per species). On each individual, three fully expanded, healthy leaves located on the first lateral branch at the base of the trunk were marked and measured (leaves = three per individual). This yielded in a total of nine measurements per species (measurements = nine). The same protocol was applied to the associated coffee plants under the canopy of each tree species, sampling three individuals per variety and three leaves attached to plagiotropic branches per coffee bush. For statistical analysis, data from the three leaves from the same individual were averaged to obtain a single representative value per individual ($n = 3$ individuals per species/variety). These averages were subsequently used for comparative analyses between species and varieties.
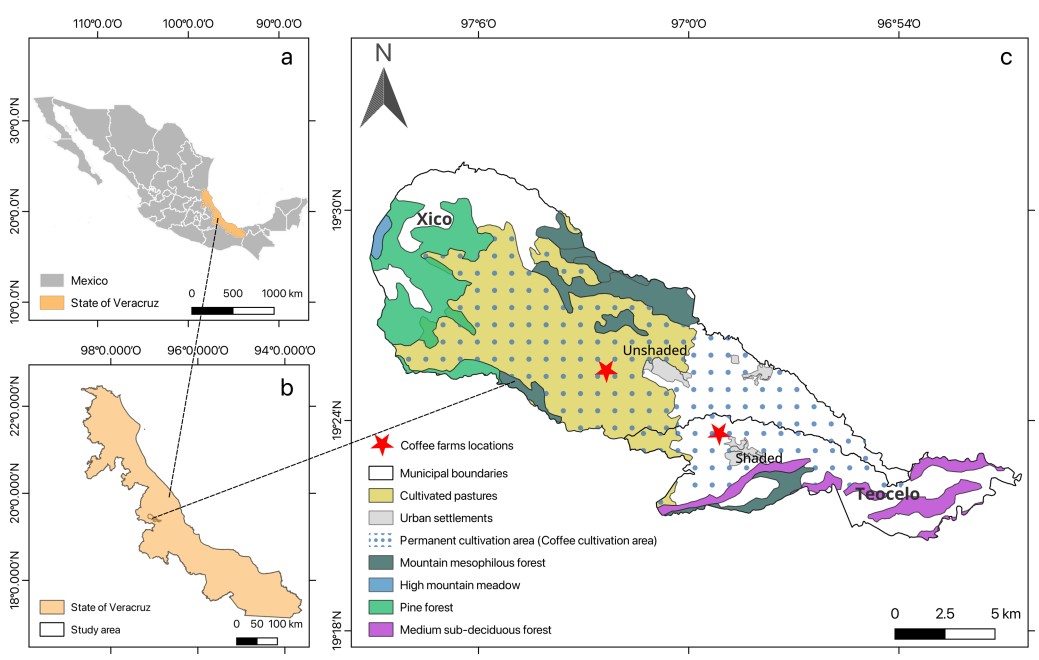

**Figure 1** **The study area location.** (A) Mexico; (B) the state of Veracruz; and (C) the municipalities with the coffee farms where the *in-situ* tests were conducted.

## Dendrometric parameters of shade trees and carbon stocks determination

Measurements were made of the total height and the DBH of three mature and reproductive individuals of each of the seven shade trees. Using these morphometric parameters, the available allometric equations were applied to calculate each tree's above ground biomass or AGB (Table 1). The allometric equations used to calculate the AGBs for *Ha* and *Pg* were at species level, at genus level for *Inga* spp. (*Ii*, *Iv* and *Ip*) and for *Erythrin* a sp. (*Ea*), and at tropical forest level for *Ps* (*Rojas-García et al., 2015*; *Ortiz-Ceballos et al., 2020*) (Table 1). AGB was converted to biomass carbon stock (CS) by multiplying by 0.47, representing the standard carbon fraction in tree biomass (*IPCC, 2006*). CS reflects both tree's ability to grow new cells and its carbon storage potential (*IPCC, 2022*).

Due to variation in tree age, 10-year normalised CS were obtained by a simple proportionality rule for each tree species. This normalisation enables standardised comparison of carbon storage potential while recognising that species-specific traits, such as shade tolerance and wood density, may influence long-term sequestration rates. Finally, we extrapolated hectare-scale carbon storage using the tree density found in the literature (references in Table 2).

## Chlorophyll fluorescence, gas exchange analyses of shade trees and coffee plants

*In-situ* measurements were performed on leaves from the seven tree species and associated-coffee plants located under the canopy of the seven tree species, which correspond to five varieties of Arabica coffee (Oa, G, Cr, T, and Ca) using a portable infrared gas analyser

Cabrera-Santos et al. (2025), *PeerJ*, DOI 10.7717/peerj.20255

**Table 1 Dendrometric parameters of shade trees.** According with their carbon stock in biomass at 10 years, species are arranged descending. The average of three replicates is presented.

| Tree Species | DBH[1] (m) | Height (m) | Age (years) | Allometric equation | AGB[2] (kg tree$^{-1}$) | Carbon stock[3] (kg tree$^{-1}$) | 10-years carbon stock (kg tree$^{-1}$) |
|---|---|---|---|---|---|---|---|
| *Persea schiedeana (Ps)* | 0.35 ± 0.05 | 14.33 ± 2.08 | 10.00 ± 4.58 | Exp((−3.1141)+((0.9719)*(Ln(DBH^2*H)) | 585.93 ± 124.85 | 275.38 ± 58.68 | 275.38 ± 58.68 |
| *Erythrina Americana (Ea)* | 0.58 ± 0.05 | 11.67 ± 2.02 | 30.00 ± 0.00 | [0.3700]*[DBH^1.9600] | 1,054.91 ± 175.26 | 495.81 ± 82.37 | 165.27 ± 27.46 |
| *Inga inicuil (Ii)* | 0.24 ± 0.00 | 12.00 ± 3.46 | 10.00 ± 0.00 | [Exp[−1.76]*[DBH^2.26]] | 223.29 ± 3.46 | 104.95 ± 1.63 | 104.95 ± 1.63 |
| *Heliocarpus appendiculatus (Ha)* | 0.37 ± 0.02 | 15.67 ± 0.58 | 20.00 ± 0.00 | [[Exp[4.9375]]*[[DBH^2]^1.0583]]*[1.14]/1000 | 331.76 ± 30.59 | 155.92 ± 14.38 | 77.96 ± 7.19 |
| *Psidium guajava (Pg)* | 0.08 ± 0.01 | 5.00 ± 0.00 | 6.00 ± 1.73 | [0.246689]*[DBH^2.24992] | 28.25 ± 6.71 | 13.28 ± 3.15 | 22.13 ± 5.25 |
| *Inga vera (Iv)* | 0.10 ± 0.02 | 5.00 ± 1.00 | 11.67 ± 2.89 | [Exp[−1.76]*[DBH^2.26]] | 30.00 ± 13.21 | 14.10 ± 6.21 | 12.09 ± 5.32 |
| *Inga punctata (Ip)* | 0.07 ± 0.00 | 4.00 ± 0.00 | 13.33 ± 5.77 | [Exp[−1.76]*[DBH^2.26]] | 15.33 ± 0.78 | 7.20 ± 0.37 | 5.40 ± 0.27 |

**Notes.**

[1] Diameter at breast height (1.30 m).

[2] Above-ground biomass (AGB).

[3] The carbon stock was determined by multiplying the calculate AGBs by the default carbon fraction of 0.47 set by the *IPCC (2006)*.

**Table 2  Carbon capture of shade tree species.** Estimated carbon stock per hectare using reported densities of the target shade trees species.

| Tree Species | ≈Trees ha$^{-1}$ | Carbon stock per hectare (Mg C ha$^{-1}$) | Reference (Trees ha$^{-1}$) |
|---|---|---|---|
| *Persea schiedeana (Ps)* | 40 | 11.0152 | *Soto-Pinto et al. (2001)* |
| *Erythrina Americana (Ea)* | 40 | 19.8324 | *Soto-Pinto et al. (2001)* and *Garza-Lau et al. (2020)* |
| *Inga inicuil (Ii)* | 200 | 20.99 | *Barradas & Fanjul (1986)* |
| *Heliocarpus appendiculatus (Ha)* | 40 | 6.2368 | *Soto-Pinto et al. (2001)* and *Romero-Alvarado et al. (2002)* |
| *Psidium guajava (Pg)* | 40 | 0.5312 | *Somarriba (1988)* and *Akter et al. (2022)* |
| *Inga vera (Iv)* | 200 | 2.82 | *Garza-Lau et al. (2020)* |
| *Inga punctata (Ip)* | 100 | 0.72 | *Valencia et al. (2014)* and *Soto-Pinto et al. (2001)* |
| **Total** | 660 | 62.1456 | |

(IRGA) LI-6400XT (Licor, Lincoln, NE, USA) equipped with a fluorometric cell. For shade trees, we assessed leaves attached to lateral branches closest to the understory layer, between 2–4 m from the ground *via* ladders and climbing equipment. These branches exhibited sympodial growth with a predominantly horizontal or obliquely orientated architecture, characteristic of species such as *Inga* spp. (Troll model), *Ea* (Champagnat model), and *Pg* (Roux model) (*Vester, 2002*; *de Reffye et al., 2008*). Coffee measurements were taken from leaves attached to plagiotropic branches at 1.30–2 m height.

Chlorophyll fluorescence analysis was performed in dark-adapted leaf tissues (30 min). Minimum (F0) and maximum fluorescence (Fm) in light-adapted tissues was measured by the saturation pulse method ($\lambda = 630$ nm, Q >7,000 $\mu$mol m$^{-2}$ s$^{-1}$, 6s). Based on these signals, variable fluorescence in the dark (Fv=Fm-F0) and quantum efficiency (Fv/Fm) were calculated (*Silva et al., 2010*; *Rakocevic et al., 2022*). Fv/Fm is frequently used to estimate the photochemical efficiency of PSII (*Niinemets & Kull, 2001*; *Lepeduš et al., 2005*; *Zavafer & Mancilla, 2021*). After that, the gas exchange parameters $CO_2$ assimilation rate, stomatal conductance, transpiration, and intercellular $CO_2$ ($C_i$) were measured in the same leaves attached to the branch. All measurements were conducted between 9:00 h and 11:00 h under controlled conditions, with the IRGA operated as an open system with a photon flux density of 1,000 $\mu$mol m$^{-2}$ s$^{-1}$, a leaf temperature of 25 °C, and an environmental air $CO_2$ concentration of $\sim$420 ppm.

Subsequent gas exchange measurements, it proceeded to estimate incident solar radiation in the understory layer between shaded and unshaded systems. PAR was measured under all tree species' canopies and sun-exposed Oro Azteca coffee plants at midday; this period sees peak solar radiation and thermal stress (*Meili et al., 2021*; *Kohl, Niether & Abdulai, 2024*).

Instantaneous leaf water-use efficiency ($WUE = A/E$) was calculated as the $CO_2$ assimilation-transpiration ratio (*Hatfield & Dold, 2019*).

## Gravimetric and analytical chemical analysis: moisture and nitrogen content of coffee leaves samples

Following immediately after chlorophyll fluorescence and leaf gas exchange measurements, the same leaves from Oro Azteca coffee plants grown under both shaded and unshaded conditions were excised at the base of the petiole. Collected leaves were stored in sealed plastic containers to prevent moisture loss until further analysis. Subsequently, the samples were processed within the next two days after field sampling. The leaves from each condition were macerated and pooled, then 0.5 g was taken for moisture and nitrogen analysis. Gravimetric methods comparing weight before and after drying determined moisture content based on the following equation:

$$H(\%) = \frac{FW - DW}{FW} \times 100. \tag{1}$$

This parameter refers to the proportion of water present in fresh leaf tissue relative to its total weight, where $H\ (\%)$ is the moisture content expressed as a percentage, $FW$ is the fresh weight, and $DW$ is the dry weight.

We employed the semimicro–Kjeldahl method (*Nelson & Sommers, 1980*; *DOF, 2019*) for digested 50 mg of dried leaf material using a digestion/distillation apparatus (Labconco®) to quantify organic nitrogen content.

## Statistical analysis

Statistical analyses were performed on the averaged data from nine measurements (described above). The assumption of normality (Shapiro–Wilk, $p > 0.01$) was verified for all datasets. All data met the assumption; thus, no transformations were required.

One-way analysis of variance (ANOVA) was used to identify significant differences ($p < 0.001$) among shade tree species for each parameter. For comparisons involving coffee varieties under different shade conditions, a one-way ANOVA was also employed ($p < 0.001$). Where ANOVA indicated significant differences, the *post-hoc* Tukey's test was applied for pairwise comparisons. Comparisons were conducted for each parameter between all shade tree species. Coffee variety comparisons were performed between the Oro Azteca variety in shaded conditions *vs* unshaded conditions. Additionally, comparisons were conducted considering only coffee varieties in shaded conditions. WUE statistical differences between shade trees and between coffee varieties were identified following the same methodology ($p < 0.001$). Comparisons of Oro Azteca moisture and nitrogen content between shaded and unshaded conditions were performed using a two-tailed Student's $t$-test ($p < 0.01$) for independent samples.

To analyse the possible clustering of the seven shade tree species and coffee varieties based on their physiological traits, a principal component analysis (PCA) was conducted on both datasets comprising all measured parameters (Fv/Fm, $CO_2$ assimilation rate, stomatal conductance, transpiration rate, and intercellular $CO_2$ concentration). PCA analysis proceeded by extracting principal components from the correlation matrix of the variables, with the selection criterion being the retention of components that collectively explained at least 75% of the total variance.

GraphPad Prism® version 9.5.1 for macOS (GraphPad Software, San Diego, CA, USA; http://www.graphpad.com) was used for all statistical analysis (accessed in January 2023).

## RESULTS

### Dendrometric parameters of shade trees and carbon stocks determination

*Ea* and *Ps* showed the highest mean AGB and CS values per tree, while *Ha* and *Ii* displayed intermediate levels, and *Iv*, *Pg*, and *Ip* had the lowest (Table 1). The 10-year normalized CS values showed similar patterns across species, where *Ps* and *Ea* had the highest carbon storage capacities, followed by lower values for the remaining species (Table 1). At the hectare scale, *Ii* achieved the highest annual CS (20.99 Mg C ha$^{-1}$), followed by *Ea* (19.83 Mg C ha$^{-1}$), *Ps* (11.01 Mg C ha$^{-1}$), and *Ha* (6.23 Mg C ha$^{-1}$), with *Iv*, *Ip* and *Pg* showing the lowest values (0.72 and 0.53 Mg C ha$^{-1}$, respectively) (Table 2).

### Chlorophyll fluorescence, gas exchange and PCA of shade trees and coffee plants

The Fv/Fm values of shade tree species (range 0.74–0.81) and coffee plants (0.73–0.80) showed consistent trends, with no significant differences observed (Table S1). Similarly, no significant differences were found between shaded and unshaded Oa or among other coffee varieties under shaded conditions (Table S2).

$CO_2$ assimilation rates differed significantly among species ($F_{(6,56)} = 118.3$, $p < 0.001$), with highest values in *Ea* (8.37 $\pm$ 0.90 µmol $CO_2$m$^2$/s), *Pg* (9.18 $\pm$ 1.00), and *Ha* (7.99 $\pm$ 1.16), and lowest in *Ps* (2.45 $\pm$ 0.40), *Iv* (2.21 $\pm$ 1.19), and *Ii* (1.67 $\pm$ 0.38) (Fig. 2B, Table S1). For Oa coffee, shaded plants showed higher rates than unshaded ($F_{(3,32)} = 54.42$, $p < 0.001$), with *Ip*-Oa (5.68 $\pm$ 0.65) and *Ha*-Oa (5.76 $\pm$ 0.91) exhibiting the highest values among shaded varieties ($F_{(6,56)} = 16.82$, $p < 0.001$) (Figs. 3A, 4A, Table S2).

Transpiration rates varied significantly ($F_{(6,56)} = 65.43$, $p < 0.001$), with highest values in *Pg* (2.59 $\pm$ 0.22 mmol $H_2O$ m$^2$/s) and *Ha* (2.34 $\pm$ 0.33), and lowest in *Ip* (1.15 $\pm$ 0.29), *Ea* (0.71 $\pm$ 0.19), and *Ps* (0.48 $\pm$ 0.10) (Fig. 2B, Table S1). Among Oa coffee variety, unshaded Oa (0.53 $\pm$ 0.04) and *Ip*-Oa (0.53 $\pm$ 0.30) showed highest transpiration, while *Ha*-Oa (0.30 $\pm$ 0.18) and *Ps*-Oa (0.23 $\pm$ 0.02) showed lowest ($F_{(2,32)} = 6.488$, $p < 0.001$) (Fig. 3B, Table S2). Among shaded coffee varieties, *Ii*-G showed the highest transpiration rate (1.02 $\pm$ 0.25) while *Pg*-Ca the lowest (0.27 $\pm$ 0.06) (Fig. 4B, Table S2).

Stomatal conductance followed similar patterns, with highest values in *Pg* (266.88 $\pm$ 29.01 mmol $H_2O$ m$^2$/s), *Iv* (245.44 $\pm$ 85.25), and *Ha* (224.88 $\pm$ 35.51), and lowest in *Ps* (26.57 $\pm$ 12.98) with significant differences between them ($F_{(6,56)} = 19.33$, $p < 0.001$) (Fig. 2C, Table S1). For the Oa cultivar, stomatal conductance varied significantly between unshaded conditions and shaded *Ha*-Oa and *Ps*-Oa ($F_{(3,32)} = 8.697$, $p < 0.001$) (Fig. 3C, Table S2). For shaded coffee, *Iv*-Cr (145.70 $\pm$ 67.86) and *Ii*-G (100.17 $\pm$ 34.05) showed highest conductance, while *Pg*-Ca (37.76 $\pm$ 24.04), *Ip*-Oa (31.65 $\pm$ 11.74), and *Ha*-Oa (15.60 $\pm$ 4.06) showed lowest ($F_{(6,56)} = 24.02$, $p < 0.001$) (Fig. 4C, Table S2).

$C_i$ values among shade trees exhibited notable variations, with the highest values observed in *Iv* (444 $\pm$ 17.34 µmol $CO_2$/mol), *Ha* (427.44 $\pm$ 12.97), and *Ii* (419.78 $\pm$ 41.88) and

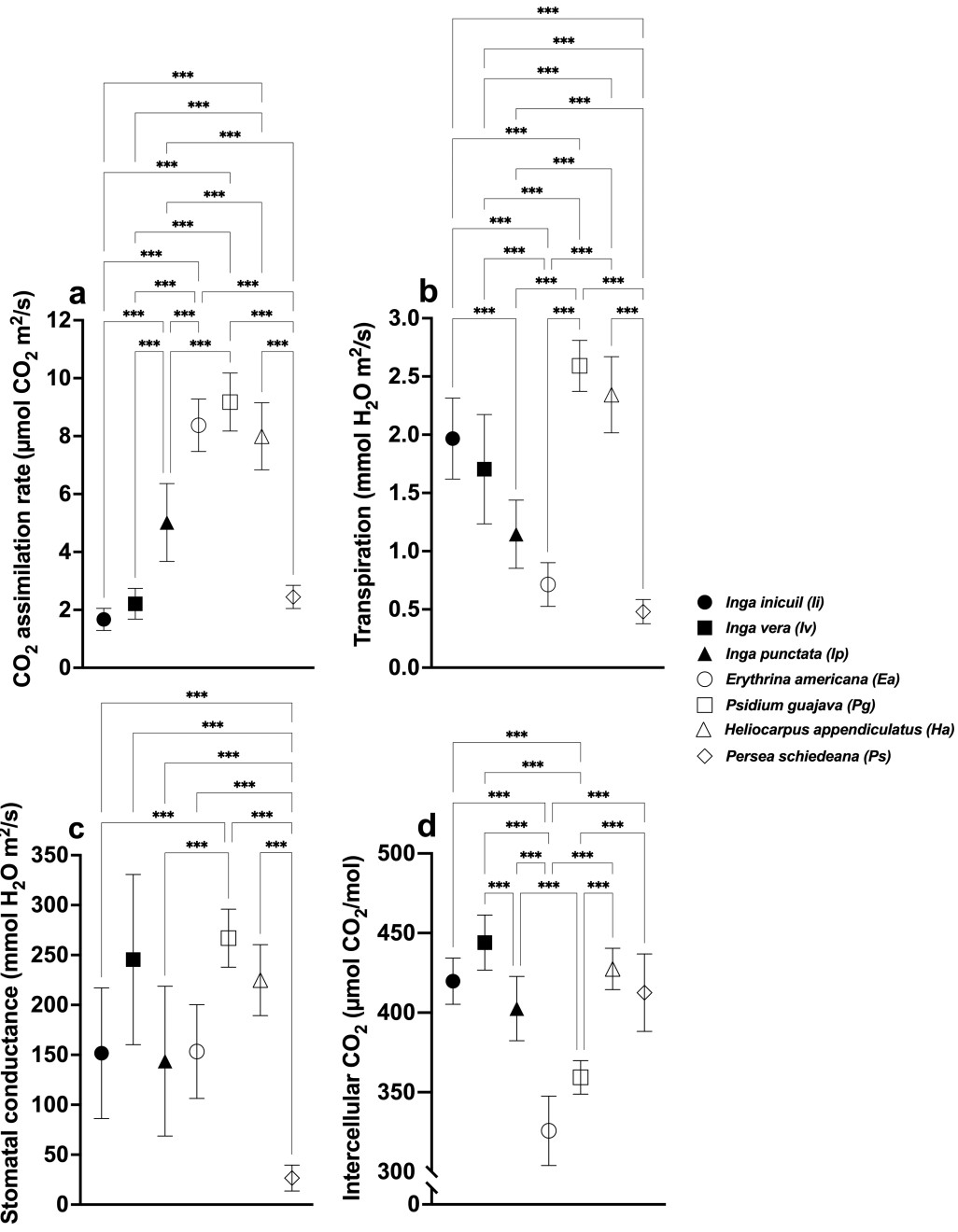

**Figure 2  Stomatal parameters of shade trees.** (A) $CO_2$ assimilation rate; (B) Transpiration rate; (C) Stomatal conductance; and (D) Intercellular $CO_2$. Mean ± SD ($n = 9$). One-way ANOVA and a *post-hoc* Tukey's test were used to identify significant differences. Comparisons with a *p* values ≤ 0.001 (***) are depicted.

lower values in *Pg* (359.33 ± 10.54) and *Ea* (325.78 ± 21.76), with significant differences between them ($F_{(6,56)} = 48.22$, $p < 0.001$) (Fig. 2D, Table S1). For the Oa variety, $C_i$ values were influenced by shaded and unshaded conditions, with *Ip*-Oa showing the

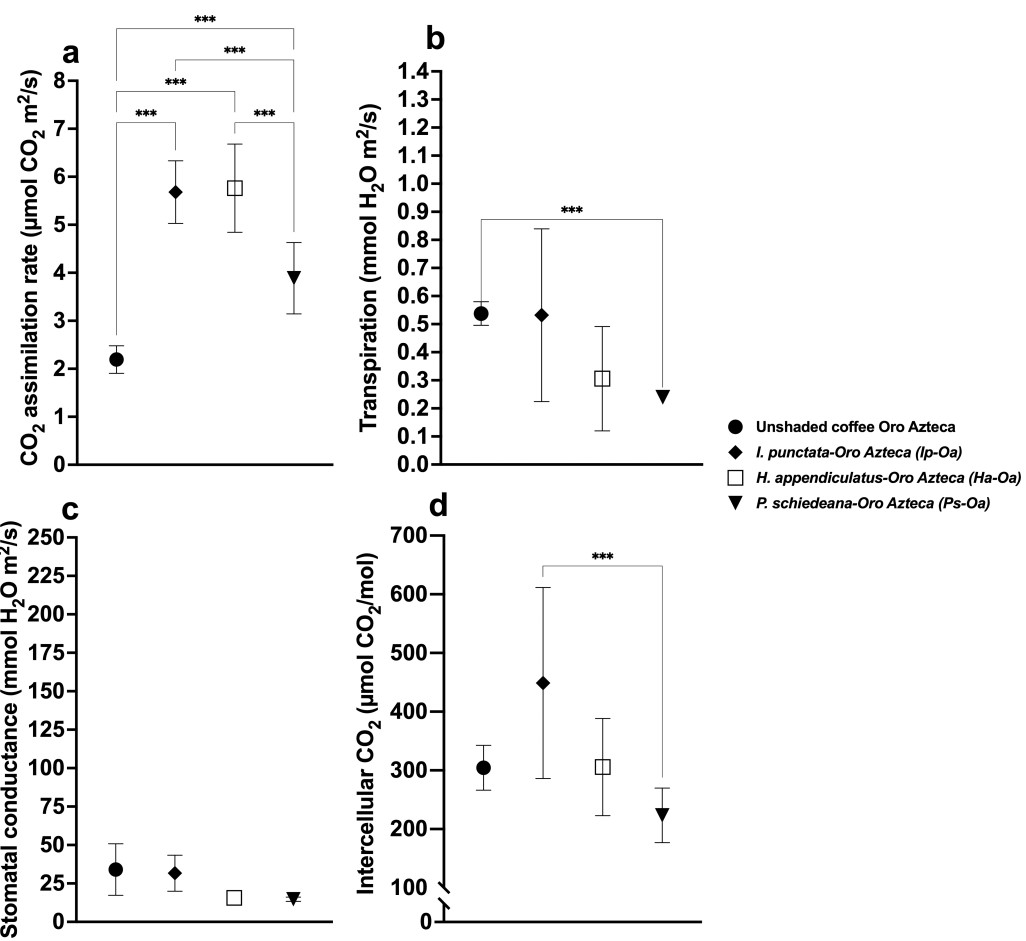

**Figure 3** **Stomatal parameters of unshaded and shaded coffee plants of the Oro Azteca variety.** Shade tree species-coffee associations are indicated in each case. (A) $CO_2$ assimilation rate; (B) Transpiration rate; (C) Stomatal conductance; and (D) Intercellular $CO_2$. Mean $\pm$ SD ($n = 9$). One-way ANOVA and a *post-hoc* Tukey's test were used to identify significant differences. Comparisons with a $p$ value $\leq 0.001$ (\*\*\*) are depicted.

highest values ($448.77 \pm 82.59$) and *Ps*-Oa the lowest ($223.44 \pm 46.47$) ($F_{(6,56)} = 48.22$, $p < 0.001$) (Fig. 3D, Table S2). Among shaded coffee varieties, *Ip*-Oa ($448.77 \pm 82.59$), *Iv*-Cr ($443.33 \pm 162.69$), and *Ii*-G ($410.55 \pm 26.82$) demonstrated higher $C_i$ values, while *Ea*-T ($284.88 \pm 34.46$), *Pg*-Ca ($284.88 \pm 34.46$), *Ha*-Oa, ($305.66 \pm 82.79$) and *Ps*-Oa ($223.44 \pm 46.47$) showed lower values, showing significant differences between them ($F_{(6,56)} = 11.89$, $p < 0.001$) (Fig. 4D, Table S2).

PAR levels differed significantly among shade tree species ($F_{(6,56)} = 64.15$, $p < 0.001$), ranging from $23.3 \pm 4.06$ $\mu$mol m$^{-2}$ s$^{-1}$ (*Ps*) to $92.2 \pm 6.66$ (*Ea*) (Table S1). For Oa coffee variety, unshaded coffee received $1,427 \pm 124.0$ $\mu$mol m$^{-2}$ s$^{-1}$ *versus* $8.44$ (*Ps*-Oa)–$25.11$ (*Ip*-Oa) under shade ($F_{(6,56)} = 47.68$, $p < 0.001$) (Table S2). Among shaded coffee varieties, *Iv*-Cr ($47.1 \pm 9.18$) and *Pg*-Ca ($32.2 \pm 5.38$) displayed the highest PAR values, while *Ps*-Oa

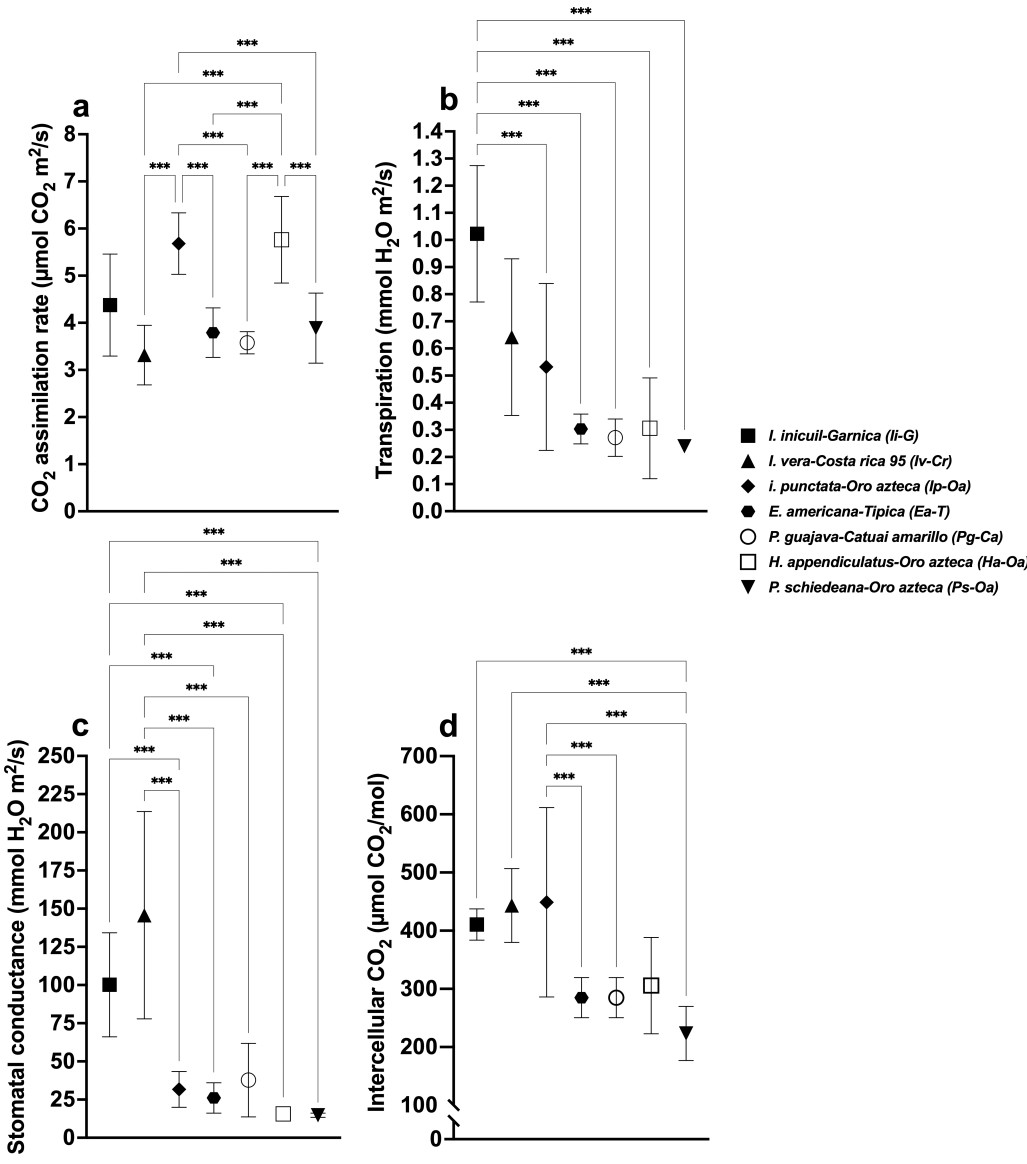

**Figure 4  Stomatal parameters of the different *Coffea arabica* varieties in shaded condition.** Shade tree species-coffee variety associations are indicated in each case. (A) $CO_2$ assimilation rate; (B) Transpiration rate; (C) Stomatal conductance; and (D) Intercellular $CO_2$. Mean $\pm$ SD ($n = 9$). One-way ANOVA and a *post-hoc* Tukey's test were used to identify significant differences. Comparisons with a $p$ value $\leq 0.001$ (***) are depicted.

$(8.44 \pm 1.08)$ and *Ha*-Oa $(14.8 \pm 6.64)$ showed the lowest measurements, with significant differences between conditions ($F_{(6,56)} = 47.68$, $p < 0.001$) (Table S2).

WUE values showed significant variation among the seven tree species, ranging from $0.83 \pm 0.15$ µmol $CO_2$/mmol $H_2O$ (*Ii*) to $16.92 \pm 11.05$ µmol $CO_2$/mmol $H_2O$ (*Ea*) ($F_{(6,56)} = 5.810$; $p < 0.001$) (Fig. 5A, Table S1). For the Oa coffee variety, WUE values ranged from $4.61 \pm 10.06$ µmol $CO_2$/mmol $H_2O$ in unshaded conditions to $21.62 \pm 11.27$ µmol $CO_2$/mmol $H_2O$ (*Ha*-Oa) ($F_{(3,32)} = 8.538$; $p < 0.001$) (Fig. 5B, Table S2). Among
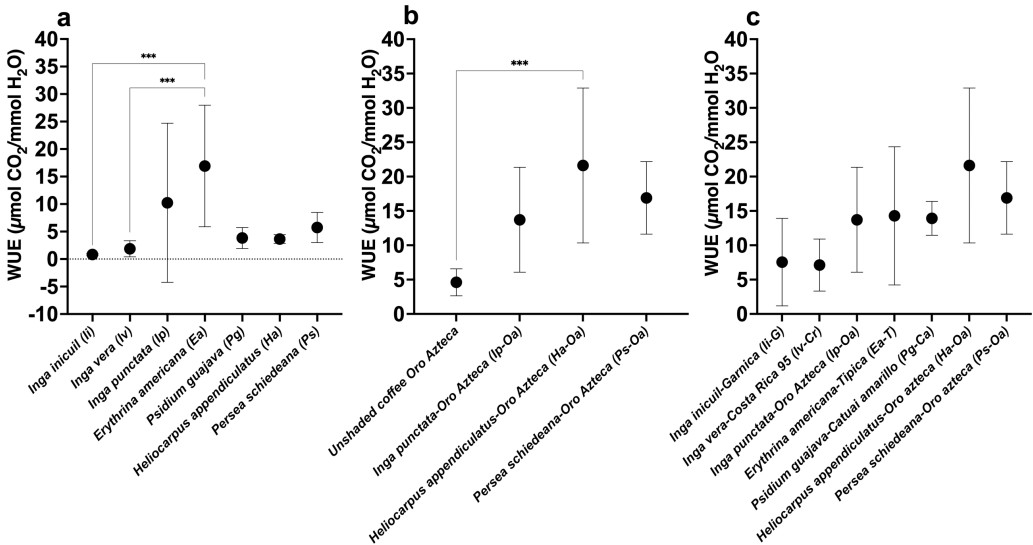

**Figure 5** **Instantaneous leaf water-use efficiency (WUE) of shade trees and coffee varieties in unshaded and shaded conditions.** (A) Shade tree species; (B) Oro Azteca coffee variety; and (C) Shaded Arabica coffee varieties. Mean $\pm$ SD ($n = 9$). One-way ANOVA and a *post-hoc* Tukey's test were used to identify significant differences. Comparisons with a $p$ value $\leq 0.001$ (***) are depicted.

shaded Arabica cultivars, values varied from $7.12 \pm 3.78$ µmol $CO_2$/mmol $H_2O$ (*Iv*-Cr) to $21.62 \pm 11.27$ µmol $CO_2$/mmol $H_2O$ (*Ha*-Oa) (Fig. 5C, Table S2); no statistically significant differences were observed.

For the PCA for the seven shade tree species, PC1 accounted for 43.57% of the variance (eigenvalue = 2.178), while PC2 contributed 33.43% (eigenvalue = 1.671), cumulatively explaining 77% of the dataset variability (Table S3). In contrast, the PCA for the coffee varieties (including Oro Azteca) under shaded and unshaded conditions revealed a stronger influence of PC1, which explained 49.04% of the variance (eigenvalue = 2.452), with PC2 adding 29.98% (eigenvalue = 1.499), resulting in a slightly higher cumulative variance (79.02%) (Table S4).

For the seven shade tree species, PC1 was strongly associated with stomatal conductance (loading = 0.882) and transpiration (0.866), with *Pg* and *Ps* contributed most to PC1 (42.5% and 37.1%, respectively) (Fig. 6A, Table S3). PC2 showed positive loading for $CO_2$ assimilation rate (0.594) and negative loading for intercellular $CO_2$ concentration (−0.937), with *Ea* contributing 73.9% to this axis (Fig. 6A, Table S3).

For coffee varieties, PC1 had negative loadings for stomatal conductance (−0.906), transpiration (−0.897), and intercellular $CO_2$ (−0.884) (Fig. 6B; Table S4), with *Ps*-Oa (22.0%) and *Ha*-Oa (5.9%) showing highest contributions (Table S4). PC2 showed positive loading for photochemical efficiency (Fv/Fm = 0.766) and negative loading for $CO_2$ assimilation rate (−0.884) (Fig. 6B, Table S4), with unshaded Oro Azteca contributing 39.3% (Table S4).

Correlation analyses for shade trees, stomatal conductance, and transpiration were strongly positively linked ($r = 0.825$), while $CO_2$ assimilation and intercellular $CO_2$ were

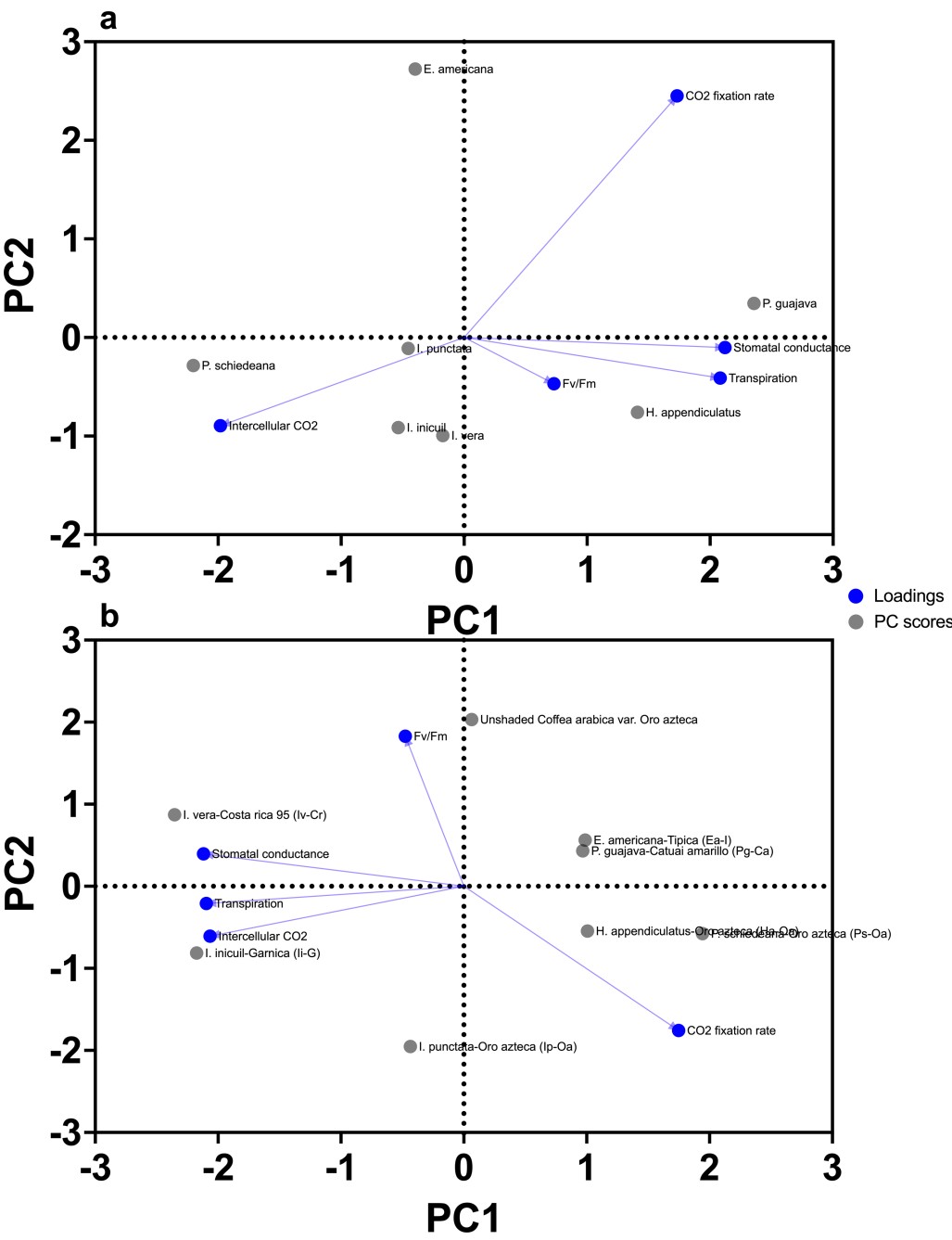

**Figure 6 Principal component analysis (PCA) of chlorophyll fluorescence and gas exchange parameters.** (A) shade tree species; and (B) Arabica coffee varieties. PC1 represents 43.57% and 49.04% of the total variation for shade trees and Arabica coffee varieties, respectively, while PC2 represents 33.43% and 29.98% of the total variation for shade trees and Arabica coffee varieties, respectively (77% and 79.02% of the total variance for shade trees and Arabica coffee varieties, respectively). Correlation of variables with PCA axes is indicated by blue solid line vectors.

negatively correlated (r = −0.689) (Table S3). For coffee varieties, stomatal conductance, transpiration, and intercellular $CO_2$ showed strong positive correlations ($r = 0.698$–$0.731$), whereas $CO_2$ assimilation and Fv/Fm were moderately negatively associated (r = −0.420) (Table S4).

### Gravimetric and analytical chemical analysis: moisture and nitrogen content of coffee leaves samples

The shaded Oa samples showed a slightly higher moisture content (55.56%) compared to the unshaded samples (53.97%) (Table S5). Also, shaded Oa samples exhibited a significantly higher nitrogen content (2.83 ± 0.06%) compared to unshaded samples (2.54 ± 0.02%) (t(4) = 7.6; $p < 0.01$) (Table S5).

## DISCUSSION

Our study provides a novel, ecophysiologically-based framework for selecting shade trees in coffee agroforestry systems by moving beyond traditional ecological or yield-based criteria. We present the first integrated analysis of *in-situ* carbon capture potential, photosynthetic performance, and water-use strategies for seven previously shortlisted native shade tree species and five *Coffea arabica* varieties in Veracruz, Mexico. By combining dendrometric assessments with controlled light and temperature measurements of chlorophyll fluorescence and leaf gas exchange, we unveil distinct functional groups and species-specific trade-offs between carbon assimilation and water conservation. Furthermore, we quantitatively demonstrate how shade mediates key leaf traits in coffee, such as nitrogen and moisture content, leading to improved water-use efficiency. Our findings showed that native shade tree species contributed differentially to carbon capture in coffee agroforestry systems, supporting our hypothesis that their physiological traits enhance both carbon sequestration and system resilience through shading and optimised resource-use efficiency. This work offers empirically supported criteria for designing climate-resilient agroforestry systems that optimise synergies between carbon sequestration, microclimate regulation, and crop physiology.

### Dendrometric parameters of shade trees and carbon stocks determination

The seven shade tree species showed distinct photosynthetic traits, growth and biomass allocation patterns, leading to inherent variability in carbon accumulation, making direct interspecific comparisons of AGB and CS unreliable (*Poorter et al., 2008*; *Chave et al., 2014*). Works such as that of *Garza-Lau et al. (2020)* showed these types of variations in agroforestry systems in the state of Veracruz. Farmers typically manage tree density through propagation programmes, though optimal balances between competition and productivity need further study. Regional factors like altitude, slope, and shade management affect phenology and influence tree traits and ecosystem services (*Bewley & Black, 1994*; *Cerda et al., 2017*; *Asanok et al., 2024*). These constraints and cultivation practices collectively determine tree densities. Coffee plants carbon sequestration in the study area, although it was not determined in our work, is highly variable (0–12 Mg ha$^{-1}$; *Valdés-Velarde et al.,*

*2022*), depending on variety and density, making extrapolation and comparison difficult, even within the same study area.

Three ecological patterns emerge when examining species contributions. First, the *Inga* genus (Fabaceae), particularly *I. inicuil*, achieves remarkable carbon capture through numerical dominance rather than individual tree performance. With 660 trees ha$^{-1}$, *I. inicuil* accounts for 33.77% of system carbon uptake, achieving 198 Mg C ha$^{-1}$, nearly double the sequestration reported for other *Inga*, *Erythrina*, and *Musaceae* species (91.64–115.5 Mg C ha$^{-1}$; *Haber, 2001*). Reported densities for *I. inicuil* range from 100 to 800 trees ha$^{-1}$, averaging 250–350 trees ha$^{-1}$ (*Barradas & Fanjul, 1986*; *Soto-Pinto et al., 2001*). After 10 years of growth, *I. inicuil* exhibited a CS of 20.9 Mg C ha$^{-1}$ at 200 trees ha$^{-1}$, exceeding values for *Inga densiflora* Benth. (24.3 Mg C ha$^{-1}$ at 400 trees ha$^{-1}$) with similar age and size parameters (*Salazar-Figueroa, 1985*; *Kursten & Burschel, 1993*). However, these values were three times lower than those reported for *I. inicuil* in Oaxaca, Mexico (64.3 Mg C ha$^{-1}$ at 164 trees ha$^{-1}$; *Hernández-Vásquez et al., 2012*; *Alessandrini et al., 2011*; *Tellez et al., 2020*), underscoring how regional factors like altitude and microclimate interact with species physiology. When comparing agroforestry systems across different states, it is important to consider that variations in management practices, soil characteristics, and climatic regimes may lead to substantial differences in carbon sequestration. Consequently, direct comparisons between regions should be interpreted with caution, as local variables shape ecosystem functioning. Although *I. punctata* had lower AGB and CS, its high density (representing 20–40% of total trees) contributed significantly to system-level carbon capture (*Soto-Pinto et al., 2001*). Incorporating both *I. punctata* and *I. vera* in agroforestry configurations may achieve carbon stocks of 91.64 Mg C ha$^{-1}$ (*Haber, 2001*).

Second, *E. americana* and *P. schiedeana* follow a quality-over-quantity approach. Their substantial trunk diameters and heights enable just 40 trees to capture carbon equivalent to 94.48% and 52.47%, respectively of the carbon captured by 200 *I. inicuil* trees. However, biological constraints, including seed dormancy in *E. americana* (*Bewley & Black, 1994*; *Bonfil-Sanders, Cajero-Lázaro & Evans, 2008*) and extensive crown-canopy development in *P. schiedeana* (*Niembro, 1992*; *Vázquez-Torres, Campos-Jiménez & Juárez-Fragoso, 2017*), naturally restrict their planting densities in managed agroforestry systems.

Third, the complementary roles of remaining species enhance system functionality. *H. appendiculatus*, representing 16–20% of tree strata in Chiapas coffee farms (*Soto-Pinto et al., 2001*; *Castillo-Capitán et al., 2014*), accounted for 10.03% of total carbon assimilation despite representing only 6% of trees in this study. *P. guajava*, which constituted 4–5% of tree density in coffee plantations (*Soto-Pinto et al., 2001*), showed the lowest CS values (6.53% of total CS at 10 years), consistent with prior findings (*Nava et al., 2009*) and attributable to its average height of 3–8 m (*Heuzé et al., 2017*). *P. guajava* can be incorporated into agroforestry systems to enhance carbon storage, particularly in leaves and roots, with a whole calculated CS ranging between 0.27 and 4.19 Mg ha–1 in 2- to 10-year-old orchards (*Naik et al., 2021*). Additionally, it provides valuable firewood and fruits (*Somarriba, 1988*; *Pascarella et al., 2000*; *Miceli-Méndez, Ferguson & Ramírez-Marcial, 2008*).

Our calculated carbon sequestration of shade tree species was highly specific. *I. inicuil* achieved system-level dominance through high density, whereas *E. americana* and *P. schiedeana* excelled through high individual tree efficiency. Optimising carbon capture therefore requires strategic species combinations that leverage these complementary ecological patterns and management practices.

## Chlorophyll fluorescence, gas exchange and PCA of shade trees and coffee plants

The physiological performance among shade trees and coffee varieties revealed distinct functional strategies shaped by interspecific variation in quantum efficiency, stomatal behaviour, and carbon capture potential. Most species maintained Fv/Fm values above the 0.75 threshold for fully functional PSII (*Genty, Briantais & Baker, 1989*; *Lepeduš et al., 2005*), indicating robust photochemical activity. This divergence underscores how intrinsic physiological traits interact with environmental conditions to determine carbon capture efficiency.

PCA identified three clusters among shade tree species, revealing distinct adaptive strategies:

### Stomatal-regulating shade tree species: P. guajava and P. schiedeana

These exhibited high stomatal conductance and transpiration, dominating PC1 (43.57% variance) with strong loadings (0.882 and 0.866, respectively) and contributions of 42.5% and 37.1%. These high values suggest a prioritisation of carbon assimilation over water loss in these species. *Nava et al. (2009)* observed a peak $CO_2$ assimilation at midday followed by an evening decline in *P. guajava* aligning with our observed $C_i$ values. Shaded conditions enhance carbon assimilation (*Idris et al., 2019*), though drought vulnerability suggests limited climate adaptability (*Maxwell & Johnson, 2000*; *Simonin, Limm & Dawson, 2012*). Conversely, *P. schiedeana* showed consistently low stomatal conductance and $CO_2$ assimilation under moderate temperatures (23.43 °C) and high humidity (70.4%). Its limited assimilation coincided with elevated $C_i$, as has been observed in *P. americana* (*Useche-Carrillo et al., 2022*), suggesting saturation kinetics where stomatal closure halts photosynthesis despite available $CO_2$ (*Sánchez-Díaz & Aguirreolea, 2008*; *Fricker & Willmer, 2012*). While *P. schiedeana*'s low carbon gain limits competitiveness under high VPD, its efficiency in stable, humid microclimates supports its agroforestry roles.

Key divergences emerge in their climate adaptations: *P. guajava*'s diurnal efficiency suits controlled-light systems, whereas *P. schiedeana*'s saturation-prone physiology demands stable humidity. Their contrasting water-use strategies and light-responsive stomata in *P. guajava* (*Idris et al., 2019*) *versus* humidity-dependent conductance in *P. schiedeana* highlight species-specific trade-offs between productivity and resilience.

### Carbon-conserving shade tree species: E. Americana

This specie exemplified a carbon-conserving strategy, dominating PC2 (33.43% variance) with a 73.9% contribution and highlighting a trade-off between $CO_2$ fixation (0.594 loading) and $C_i$ (−0.937). It exhibited high $CO_2$ assimilation rates despite suboptimal Fv/Fm values (<0.75) and lower stomatal conductance. Such traits align with observations

in other *Erythrina* species, where light-saturated photosynthesis couples with high water-use efficiency (*Nygren, 1995*; *Davis & Hidayati, 2019*). This behaviour reflects adaptation to low-VPD conditions, where elevated leaf water potential enhances stomatal efficiency (*Running, 1976*; *Dai, 2013*; *Grossiord et al., 2020*). This adaptive mechanism enhanced leaf water potential and assimilation efficiency, contrasting with *Inga* species, which maintained slower stomatal closure (*Engineer et al., 2016*; *Xu et al., 2016*). Its pioneer ecology and optimal temperature range (∼28 °C; *García-Mateos, Soto-Hernández & Vibrans, 2001*; *Palma-Garcia & Gonzales-Rebeles Islas, 2018*) support climate resilience, with reduced transpiration under high temperature/radiation enhancing water conservation. WUE analysis revealed reduced transpiration under high temperature and radiation conditions, prioritising water conservation, a trait advantageous for drought-prone agroforestry.

### Intermediate species: I. inicuil, I vera, I. punctata, and H. appendiculatus

This group showed balanced traits across both PCs. *I. inicuil*, *I. vera*, and *I. punctata* exhibited lower photosynthetic rates ($1.67 \pm 0.38$ to $5.02 \pm 1.34$ µmol $CO_2$ $m^{-2}$ $s^{-1}$) than other *Inga* species under controlled conditions (10.60–11.65 µmol $CO_2$ $m^{-2}$ $s^{-1}$; *Dos Santos Pereira et al., 2019*), likely due to differences in measurement conditions, such as mature leaves *versus* younger leaves, light intensity, and temperature All three species reduced stomatal opening under moderate temperatures (23.43 °C) and high humidity (70.4%), limiting water loss more than carbon fixation (*Shimshi & Ephrat, 1975*). Under elevated temperatures, their slower stomatal closure allowed sustained transpiration (*Engineer et al., 2016*; *Xu et al., 2016*), albeit at the cost of reduced WUE. In contrast, *H. appendiculatus* (Malvaceae), a pioneer species, showed higher assimilation (3.7–11.6 µmol $CO_2$ $m^{-2}$ $s^{-1}$) as has been seen across light regimes (*Tinoco-Ojanguren & Pearcy, 1995*), with stomatal conductance and $C_i$; positively correlated with carbon gain (*Farquhar & Sharkey, 1982*). Its plasticity stems from leaf-level adjustments, increased stomatal density under dim light and thicker leaves in high radiation (*Fetcher, Strain & Oberbauer, 1983*; *Friend, 1984*), enabled efficiency in diverse light environments (*Stegemann, Timm & Küppers, 1996*). Unlike *Inga* species, its photosynthesis was unaffected by light quality (*Tinoco-Ojanguren & Pearcy, 1995*), suggesting broader niche tolerance. Under high VPD, Inga species conserved water *via* stomatal closure (*Sinclair et al., 2017*), while *H. appendiculatus* prioritised carbon fixation, a critical trait for early succession. For agroforestry, this implies *Inga* species stabilise systems under drought, whereas *H. appendiculatus* optimises productivity in variable light.

The observed highest WUE in *I. punctata* and *E. americana*, suggests drought resilience (*Beer et al., 1998*; *Chaves, Osório & Pereira, 2004*; *Sinclair et al., 2017*), though WUE is limited by short-term measurement scales (*Medrano et al., 2007*).

Notably, coffee varieties exhibited inverse stomatal regulation patterns compared to shade trees, with PC1 loadings of −0.906 for conductance and −0.897 for transpiration, suggesting different water-use strategies under shaded conditions. Arabica coffee clustered into three groups: (1) *P. schiedeana*-Oro Azteca and *H. appendiculatus*-Oro Azteca, with high stomatal regulation and WUE but reduced $C_i$; (2) unshaded *C. arabica* var. Oro Azteca, showing light-adapted but carbon-limited photosynthesis; and (3) intermediate varieties (*I.
*inicuil*-Garnica, *I. vera*-Costa Rica 95) with balanced traits. Unshaded coffee plants showed low $CO_2$ assimilation, related to a higher temperature and lower RH regarding shaded plants. Variability in $CO_2$ assimilation rates between varieties also suggests differential sugar and starch accumulation during photosynthesis (*Riaño, 1993*; *Mosquera-Sanchez et al., 1999*).

Previously evidence suggest that unshaded coffee plants showed detrimental effects due to the effect of temperature above 25 °C on stomatal conductance and $CO_2$ assimilation, since these plants are exposed to higher radiation and VPD, causing the loss guard cell turgor and stomatal resistance (*Makino, Nakano & Mae, 1994*; *Riaño, 1993*; *Larcher, 1994*; *Roháček, 2002*). On the contrary, shaded plants exhibited more stable $CO_2$ assimilation rates, possibly linked to moderated microclimatic conditions provided by shade tree canopy.

WUE has been observed to range from 4 to 12.5 $\mu$mol $CO_2$/mmol $H_2O$ in drought-tolerant cultivars (*Reis Filho et al., 2022*), but our shaded *H. appendiculatus*-Oro Azteca association achieved 21.62 $\mu$mol $CO_2$/mmol $H_2O$, double these values. Comparative analysis of 21 genotypes revealed generally lower seasonal WUE stability (1.2–3.4 $\mu$mol $CO_2$/mmol $H_2O$; *Tezara et al., 2022*), with only unshaded Oro Azteca plants approaching our observed values.

Correlation analyses reinforced these patterns: stomatal conductance and transpiration were strongly positively linked in shade trees ($r = 0.825$) and coffee ($r = 0.698$–$0.731$), while $CO_2$ assimilation and $C_i$ were negatively correlated in trees ($r = -0.689$), highlighting a conserved carbon-water trade-off. PAR levels beneath canopies differed significantly, with *E. americana* and *Inga* species allowing higher understory PAR (365–379 $\mu$mol m$^{-2}$ s$^{-1}$) due to open canopies (*Dos Santos Pereira et al., 2019*), while *P. schiedeana* and *P. guajava* reduced irradiance by 40–60% (*Siles et al., 2012*; *Idris et al., 2019*). *H. appendiculatus* filtered ~60% of full sunlight (*Tinoco-Ojanguren & Pearcy, 1995*; *Stegemann, Timm & Küppers, 1996*), and coffee beneath dense canopies experienced PAR 30–169 times lower than unshaded conditions, mitigating photoinhibition but potentially limiting $C_3$ photosynthesis (*Roháček, 2002*; *Nava et al., 2009*). These PAR disparities highlight trade-offs between photoprotection and light availability for understory crops (*Gholipoor et al., 2010*; *Sinclair et al., 2017*).

Our results showed how PCA unveils physiological adaptations that can enhance sustainability, productivity, and climate resilience (*Hatfield & Dold, 2019*), aligning ecological and agricultural goals in water-scarce regions. Three distinct shade tree strategies were observed: *P. guajava* and *P. schiedeana* regulate stomatal gas exchange, *E. americana* conserves carbon with high WUE, and intermediate species like *Inga* species and *H. appendiculatus* balance traits. Coffee plants under shade exhibited more stable carbon assimilation than unshaded ones. A strong carbon-water trade-off was consistent across species, underscoring that optimal agroforestry systems require strategic species pairing based on these complementary physiological adaptations.

### Gravimetric and analytical chemical analysis: moisture and nitrogen content of coffee leaves samples

The lower air temperature and light intensity in shaded areas contribute to higher moisture content in coffee leaves by increasing relative humidity, which lowers the VPD (*Schwerbrock & Leuschner, 2017*). In contrast, coffee plants in unshaded conditions are subjected to environmental variables more likely to trigger plant stress responses compared to shaded plants and previous evidence on the nitrogen content of leaves has demonstrated that, in stressful situations, increased leaf nitrogen availability promotes the activation and maintenance of photoprotective systems to avert photooxidation (*Fahl et al., 1994*; *Ramalho et al., 2000*), which allows shaded leaves to adapt more efficiently to different irradiation conditions than fully sun-exposed leaves (*Araujo et al., 2008*).

Gravimetric and chemical analysis suggests that shaded conditions could have promoted higher leaf moisture content due to increased humidity and lower VPD. Conversely, unshaded coffee plants exhibited traits suggesting environmental stress, with nitrogen likely allocated to support photoprotection mechanisms. These findings highlight a physiological divergence in coffee leaf characteristics mediated by shade management practices.

Some limitations of this study should be considered. The measurements were conducted during the cool season to ensure environmental homogeneity; therefore, physiological responses during warmer or drier periods remain to be investigated. Furthermore, while we infer the effects of shade on microclimate, direct measurements of air temperature, relative humidity, and VPD at the leaf level were not concurrently recorded. Future longitudinal studies across seasons, incorporating continuous microclimate monitoring alongside physiological measurements, would provide a more comprehensive understanding of plant responses to dynamic environmental conditions. Establishing permanent sample plots would also allow for tracking long-term carbon storage and physiological acclimation. Despite these limitations, our study offers a robust snapshot of the physiological mechanisms governing species performance and provides critical insights for the immediate selection and management of shade trees.

## CONCLUSIONS

Our findings support the hypothesis that the physiological traits of native shade trees enhance carbon capture and system resilience. We demonstrate that this is mediated through distinct strategies: *E. americana* achieves high individual carbon storage and WUE, supporting resilience in drought-prone scenarios, while *I. inicuil* provides rapid, density-driven carbon capture. Remarkably, shade provision mitigated microclimatic stress for coffee, as evidenced by the higher WUE of shaded Oro Azteca variety plants. Thus, strategic species selection, combining high-carbon species like *E. americana* and *P. schiedeana* with high-tree-density *Inga* species, directly enhances carbon sequestration capacity and improves agroforestry resilience under climate change by optimising resource-use efficiency and reducing physiological stress.

Furthermore, the functional clustering of species reveals a spectrum of adaptive strategies that underline this resilience. For instance, pairing deep-shading species with drought-tolerant ones to mitigate both high radiation and water deficit.

Importantly, we also demonstrate that these shade tree traits mediate improved resilience in coffee, as hypothesised. The significantly higher WUE and stable carbon assimilation observed in shaded *C. arabica* var. Oro Azteca, particularly under *H. appendiculatus* and *P. schiedeana*, provide direct evidence of stress reduction *via* moderated microclimates. This contrasts with the limitations seen in unshaded plants, confirming that shading is a key mechanism for optimising resource-use efficiency in associated crops.

Consequently, our findings suggest a shift in agroforestry planning from a focus solely on tree density to a trait-based selection framework. The key criteria for species selection must integrate physiological performance data, specifically carbon assimilation rates, WUE, and shade density, to match species to local environmental constraints and production objectives. This relationship is essential for maximising the documented benefits of tree-crop relationships, ultimately supporting more sustainable and climate-resilient coffee production systems.

## ACKNOWLEDGEMENTS

We thank the staff and coffee smallholders in Xico and Teocelo for their support in field work.

### Funding

This research is part of the project "Enhancing carbon sequestration and improving livelihoods in shade-grown coffee plantations in the State of Veracruz, Mexico", led by the Royal Botanical Garden of Kew, in collaboration with the Faculty of Higher Studies Iztacala of the National Autonomous University of Mexico, in partnership with Pronatura Veracruz A.C. It was funded by UK PACT Mexico and has the support of the Embassy of the United Kingdom in Mexico. It is also funded by Technological Research and Innovation Support Program (PAPIIT), UNAM No. IG200323, and the Aldama Foundation. The funders had no role in study design, data collection and analysis, decision to publish, or preparation of the manuscript.

### Grant Disclosures

The following grant information was disclosed by the authors:
The Royal Botanical Garden of Kew.
The Faculty of Higher Studies Iztacala of the National Autonomous University of Mexico.
UK PACT Mexico.
The Embassy of the United Kingdom in Mexico.
Technological Research and Innovation Support Program (PAPIIT): UNAM No. IG200323.
The Aldama Foundation.

### Competing Interests

The authors declare there are no competing interests.

## Author Contributions

- Daniel Cabrera-Santos conceived and designed the sampling strategy, performed the field measurements, analyzed the data, prepared figures and/or tables, authored or reviewed drafts of the article, and approved the final draft.
- Patricia Dávila conceived and designed the sampling strategy, authored or reviewed drafts of the article, and approved the final draft.
- Isela Rodríguez-Arévalo performed the sampling strategy, authored or reviewed drafts of the article, and approved the final draft.
- Anabel Ruiz-Flores conceived and designed the sampling strategy, performed the field measurements, authored or reviewed drafts of the article, and approved the final draft.
- Josefina Vázquez-Medrano conceived and designed the sampling strategy, performed the field measurements, prepared figures and/or tables, authored or reviewed drafts of the article, and approved the final draft.
- Salvador Sampayo-Maldonado conceived and designed the sampling strategy, performed the field measurements, analyzed the data, prepared figures and/or tables, and approved the final draft.
- Cesar Ordoñez-Salanueva performed the sampling strategy, analyzed the data, authored or reviewed drafts of the article, and approved the final draft.
- Maraeva Gianella analyzed the data, authored or reviewed drafts of the article, and approved the final draft.
- Elizabeth Bell analyzed the data, authored or reviewed drafts of the article, and approved the final draft.
- María Toledo-Garibaldi analyzed the data, prepared figures and/or tables, authored or reviewed drafts of the article, and approved the final draft.
- Robert Manson analyzed the data, prepared figures and/or tables, authored or reviewed drafts of the article, and approved the final draft.
- Flor G. Vázquez-Corzas performed the sampling strategy, authored or reviewed drafts of the article, and approved the final draft.
- Jazmin Cobos-Silva performed the sampling strategy, authored or reviewed drafts of the article, and approved the final draft.
- Cesar Mateo Flores Ortiz conceived and designed the sampling strategy, performed the field measurements, authored or reviewed drafts of the article, and approved the final draft.
- Tiziana Ulian conceived and designed the experiments, analyzed the data, authored or reviewed drafts of the article, and approved the final draft.

## Data Availability

The AGB parameters of the tree species, the photosynthetic parameters of the tree and coffee species studied, and raw data obtained by PCA analysis are available in the Supplementary Files.

## Supplemental Information

Supplemental information for this article can be found online at http://dx.doi.org/10.7717/peerj.20255#supplemental-information.

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
