# Peer review of "Carbon capture, photosynthesis, and leaf gas exchange of shade tree species and Arabica coffee varieties in coffee agroforestry systems in Veracruz state, Mexico"

_PeerJ, doi:10.7717/peerj.20255_

## Round 0.1 · original submission · Major Revisions

· Academic Editor

Major Revisions

Dear Dr Ortiz,

Your article entitled "Carbon capture, photosynthesis, and leaf gas exchange of shade tree species and Arabica coffee varieties in coffee agroforestry systems in Veracruz state, Mexico" has been reviewed by two independent experts. Both agreed that the paper may be published in PeerJ, but it must first be thoroughly revised. Please review the reviewers' comments and address all of them accordingly.

With best regards,

·

Basic reporting

Line 165
Materials and Methods
- Regarding the coffee plants and trees, what was the density at the time of the study?
What was the age of the coffee plant?
Line 200
What was the criterion for measuring physiological activity at midday?
Line 205
Regarding the plagiotropic branch evaluated, in what position on the tree was it located: lower, upper, or superior?

Experimental design

-

Validity of the findings

-

·

Basic reporting

Abstract – details in the PDF. Please give information about the PPFD above the tree canopy that is coming to the coffee plants. You are selling a frog in very general lines, and you must be more specific.

Keywords – Organize by alphabetical order and DO NOT repeat terms used in the title

Introduction – lines 44-46 This is one wrong piece someone published and copied and pasted, wrongly. Please, delete

Lines 47-48: exported from to ??? No sense of your statement.
Line 49 (and the whole text): Organize publications by year
Lines up to 65 – important for your work?
Line 68 – What is an alternative for you?
Up to line 78, you did not say a word about your work! It can be deleted!
Line 81: What is the meaning of 0.31 Pg C year-1? Related to coffee and coffee systems? If not, please, do not put everything about anything. That’s not correct. Be strict, direct, and clean.

First time a species is mentioned, write the complete scientific name and scientific authority. Even for Coffea arabica.

The introduction is very political, as a defense and justification for one project. Too much story. Dry, clean, and clear – what is known and based on all parameters measured, and what was expected. |I am tired, and only at line 130!. Here must stop the Introduction. 6000 words. 1000-1500 for introduction, 1500 for M&M, 1500 for Results, 1500-2000 for discussion and Conclusions. I did not read only what the producer ate for lunch.

Line 130: Meaning of the early stages? Of trees, knowing about intensive removing and incorporation into the lignin mass? What are you talking about?

Nothing was introduced that was researched. Only very slight C storage! Because the coffee C footprint is not positive (search for the paper of Ramalho, 2014, I think). Introduce coffee responses as leaf gas exchanges, Chl fluorescence, and WUE to shade and to elevated/low temperatures and light.

Lines 153-163: Define stages, ages, species… Be direct about your results.

M&M
Species must be nominated and described, and Table 1 can be deleted.
Lines 174-179: Explain - tree age, density of planting, coffee age, density, and arrangements.
Give the complete description here and for all devices - model, brand, city, state, country.
Organize the text so that overlap and repetition do not occur. Too much, tired of reading chaotic text, and most importantly, as age, arrangement, and density had not been defined.
Line 196: Due to variation in tree age! Not defined!
Line 200: WHY DID YOU MEASURE AT MIDDAY WHEN PHOTOSYNTHESIS IS THE LOWEST?
Lines 204-205: Three stomatal and three photosynthetic parameters were measured on fully developed shade tree leaves that were located on initial plagiotropic branches of three individuals’ trees and coffee bushes per species or coffee variety.
Do all tree species have plagiotropic branches, like coffee? All have the development following the Roux model????
Dirty text, not clean at all; the Methodology is wrong. Who is measuring leaf gas exchanges and Chl fluorescence at midday? And why???? Only to understand the diminution at midday, compared to morning hours.
The explanation of how leaf gas exchanges and Chl fluorescence were measured at the same time is missing. There is one lag time, normally. How did you attain leaves in trees?????? What was the plant height? How was it measured? You did not make a good story, an explicit experimental design…
Line 235 – PAR was similar… That’s impossible, all my life I have been measuring PPFD….
Line 249- How did you collect when and from which part of the plants?
Here I stop and judge that the work was not scientific enough, and without profound restructuring, the information about the facts is not available.
Nothing is explained; nothing is clear. How different ages and arrangements can be compared. How were they collected, when, and where? The leaf photosynthesis of trees 14-15 m in height had been measured, how, where, and when? Where in the crown? How did you attain?
Chl fluorescence is NOT photosynthesis…
Cluster analyses were made how?
The number of repetitions is missing.

My judgment is to remake the manuscript with very focused data, because the most important thing here is in the Supplementary Material. Give 8 tables + figures with sense and certainly what you are talking about. This can be a valuable manuscript, but it has no experimental design or correct methodology, or even knowledge about results…

Experimental design

Zero value.

Validity of the findings

Low.

---

## Round 0.2 · Major Revisions

· Academic Editor

Major Revisions

Dear Authors,

Your work has been re-evaluated by an independent expert. They appreciated your effort in making the revisions, but still have some concerns. Kindly review these comments and respond to them accordingly.

With best regards,

·

Basic reporting

The manuscript was changed, but I did not see any marks or tracing. Your responses that you made or discussed with me in the response letter are difficult to find in the text. Normally everything MUST be found and all changed lines MUST be declared. I am waiting for previously demanded changes and herein examined, to see.

The English and sense of phrases must be checked.

Unfortunately, the abstract has no reasonable sense. You are declaring physiological measurements made in laboratory (in abstract), while in M&M they are all in situ. Detailed in pdf.

Experimental design

Still not completely clear, but it was improved. Some questions are in pdf.

You made serious errors about fluorescence measurements with the PPFD intensity, and those measurements, due to errors effectuated, MUST be withdrawn from manuscript. Fm is measured at > 6000 micromol m-2 s-1.

How did you climb to 4m of height???

Validity of the findings

Due to exposed in the previous, the authors must review all fundings and results.

Additional comments

The manuscript was improved, but please, write all changes in other color and revise the M&M. Figures are not of clear and clean resolution and as large, I would prefer to put two in each line. In Results you are giving some numbers of p values as = 0.001 (GIV the exact value, please) and <0.033. What is this? Please, give the exact p-values, always. In figures you are fiving ** and *** all < 0.0001. There must exist a difference.

---

## Round 0.3 · Major Revisions

· Academic Editor

Major Revisions

Dear authors,

One of the reviewers still has serious concerns. Please review these comments carefully and revise the manuscript accordingly.

With best regards,

·

Basic reporting

Please, after three times of suffering...

OK, manuscript can be useful if the ideas are clear and clean.

Keywords are not reflecting, at all, the results, only author's dreams.

Structure is still problematic; discussion is repeating results separately of discussion development based on previous publications. The conclusion about everything is missing at the end of each paragraph. Conclusions must respond to hypotheses (existing???), not to exhaustively repeat the results.

Experimental design

Complicated, improved in explanations, but even, authors are still not understanding proper design:

“All measurements were assayed in triplicate. Collected data from each individual were averaged per tree and per tree species or coffee variety (n=9).” (lines 264-265)

How does 3 become 9? And even more, when they are using average? Or how can you average various species??? If three species were analyzed, they are three species. This is not right at all! If averaged becomes 1…

I am not sure if authors think (design is not clean at all with trees of different ages, space and time)...

Validity of the findings

Novelty OK, but design is not clean at all...

Additional comments

The manuscript was improved. Some points still need improvements, such as the sense of some phrases or some terms. Details are in pdf.
Abstract, lines 60-61: PAR levels under shade tree species were lower than in unshaded coffee, suggesting long-term productivity and photoprotection benefits. (no logics or deduction, those are not related conditions).
Keywords are not useful. The manuscript must declare the exact parameters worked in results, as photosynthesis, stomatal conductance… not desires and dreams.
Line 153: Photosynthetic Photon Flux Density is not a personal name, and you can use small case letters.
Line 199: ONLY 1.7 mm of precipitations per year? There must be some error here.
Line 236 and forward: Please, substitute ‘CO2 fixation rate’ with ‘CO2 assimilation rate’. The assimilated was not fixed…
Statistical analysis continues to be chaotic:
1. Authors wrote: “All measurements were assayed in triplicate. Collected data from each individual were averaged per tree and per tree species or coffee variety (n=9).” (lines 264-265)
How does 3 become 9? And even more, when averaged? Or how can you average various species??? If three species were analyzed, they are three species. This is not right at all! If averaged becomes 1….
2. How “two-tailed t-test (t(4) = 7.6, p = 0.002)” (Line 275) was fixed on 0.002?????
Results were improved, but still, in many cases we do not find figure or table where to find the statements. Please, always in the beginning of paragraph, or changing the figure/table, indicate where the reader must look. Such situation exists at many places…
Discussion, Lines 408-410: “Overall, the findings highlight the diverse nature of agroforestry systems and the improved microclimatic conditions provided by tree shade.”
You did not show such conditions! Only ecophysiology and indirectly, climate changing. You did not measure the difference in temperature, PPFD, or relative humidity, soil water content…
Discussion is repeating results, did not use results to develop discussion and give always some ‘small’ conclusion.
Conclusions: Again, repetition of results. PLEASE, only respond to your hypotheses!!! Please…

---

## Round 0.4 · accepted · Accept

· Academic Editor

Accept

Dear Authors,

The team of editors has re-evaluated your work. We have concluded that you have addressed all of the reviewers’ comments, and the paper can be published in its current version. Congratulations!

For instance, please spell out water-use efficiency (WUE) in the conclusions.